# Regulatory Light Chains in Cardiac Development and Disease

**DOI:** 10.3390/ijms22094351

**Published:** 2021-04-21

**Authors:** Kasturi Markandran, Jane Wenjin Poh, Michael A. Ferenczi, Christine Cheung

**Affiliations:** 1Lee Kong Chian School of Medicine, Experimental Medicine Building, 59 Nanyang Drive, Nanyang Technological University, Singapore 636921, Singapore; KAST0008@e.ntu.edu.sg (K.M.); JANE0019@e.ntu.edu.sg (J.W.P.); m.ferenczi@ntu.edu.sg (M.A.F.); 2Brunel Medical School, Brunel University London, Kingston Lane, Uxbridge UB8 3PH, UK; 3Institute of Molecular and Cell Biology, 61 Biopolis Drive, Proteos, Singapore 138673, Singapore

**Keywords:** regulatory light chains, myosin regulatory light chain 2, RLC phosphorylation, cardiogenesis, cardiac muscle contractility

## Abstract

The role of regulatory light chains (RLCs) in cardiac muscle function has been elucidated progressively over the past decade. The RLCs are among the earliest expressed markers during cardiogenesis and persist through adulthood. Failing hearts have shown reduced RLC phosphorylation levels and that restoring baseline levels of RLC phosphorylation is necessary for generating optimal force of muscle contraction. The signalling mechanisms triggering changes in RLC phosphorylation levels during disease progression remain elusive. Uncovering this information may provide insights for better management of heart failure patients. Given the cardiac chamber-specific expression of RLC isoforms, ventricular RLCs have facilitated the identification of mature ventricular cardiomyocytes, opening up possibilities of regenerative medicine. This review consolidates the standing of RLCs in cardiac development and disease and highlights knowledge gaps and potential therapeutic advancements in targeting RLCs.

## 1. Introduction

Myosin regulatory light chains (RLCs) play a primary role in striated muscle contraction by regulating the movement of myosin head molecules for cross-bridge formation. The phosphorylation of RLC residues promotes the movement of myosin heads towards the actin filaments. A notable feature of RLCs is that they can be disengaged and reconstituted at their designated locations on the myosin molecule in vitro [1]. Such spatial manipulation of RLCs can be further exploited in vivo in order to better understand the mechanisms of RLCs in regulating muscular function. The RLCs also serve as useful markers for the study of cardiac development as they are among the first few cardiac markers expressed during cardiogenesis in various species. For advancement in cellular disease models and regenerative medicine, cardiac chamber-specific isoforms of RLCs may be exploited to derive pure populations of mature ventricular cardiomyocytes. This review consolidates the current understanding of RLCs in heart development and diseases, highlights knowledge gaps and suggests potential applications.

## 2. Structure of Regulatory Light Chains

The RLCs of myosin are ~20 kDa proteins present in striated muscles with isoform variability between muscle fibre types. The RLC isoforms have similar structures, but amino acid sequences are only evolutionarily conserved to some extent (Figure 1a), suggesting that RLCs may function distinctively in different species and even tissues [2,3,4]. The RLC is located within the sarcomere, a basic contractile unit of striated muscles. It comprises actin and myosin filaments interdigitating together, as well as other, but less abundant, proteins that provide structural support and regulate muscle contraction.

The RLC is positioned on the myosin heavy chain (MHC) molecule (myosin molecule) which ends with a globular head (Figure 1b). It is noncovalently bound at the neck region (head–rod junction) of the myosin molecule. Studies established that RLC stabilises the neck region by wrapping its N-terminal region around the C-terminus of the MHC molecules, while the C-terminal region also associates with the MHC molecules (Figure 1c) [5]. The RLC adopts a structure similar to calmodulin (calcium-modulated) proteins and acquires a helix–loop–helix motif. It has a cation binding site, which binds to either the Ca^2+^ or Mg^2+^ [6,7,8,9]. There is evidence that this site is occupied by Mg^2+^ ions under the relaxed state of skeletal muscles and is increasingly occupied by Ca^2+^ ions as its concentration increases in the event of muscle contraction [10]. The significance of this divalent binding site for regulation of both skeletal and cardiac muscle contraction was rejected as the transition in the type of divalent ion at the binding site is slow (~8 s^−1^) on the time-scale of contraction (~50 ms for twitch contractions) [9,11,12]. This conclusion is gathered based on in vitro experiments [9]. However, loss-of-function studies via site-directed mutagenesis showed that the divalent binding site is necessary for muscle contraction [13,14]. The RLC possesses two phosphorylatable sites, serine 14 and 15 in mouse and rat cardiac RLC, respectively, and another phosphorylatable residue, serine 15, in the human cardiac RLC [2,15]. Asparagine 14 is present in the human cardiac RLC sequence, instead of serine 14. Interestingly, asparagine 14 can be deaminated to aspartic acid (negative charge) to mimic a phosphorylated residue [16]. However, this has not been observed in vivo. The natural differences in the RLC isoform structures suggest that the roles of RLCs may be different in different locations. This may contribute to the characteristics of different fibre types—slow-twitch or fast-twitch fibres—or affect muscle energetics [17]. Thus, it is important to study the structures of various RLC isoforms and correlate them to functional parameters.

## 3. Regulatory Light Chains in Cardiac Development

The heart is the first organ to be developed during embryogenesis. The precursor of the heart is the primitive heart tube, from which atria, ventricles, outflow track and inflow track form via major morphological events, namely, cardiac looping and chamber septation [18]. Myosin regulatory light chain 2 (MLC-2) is one of the earliest cardiac-associated markers expressed in the primitive heart tubes of vertebrates [18,19]. In species such as mouse and rat, there are two MLC-2 isoforms, MLC-2a (atria specific) and MLC-2v (ventricle specific) [20,21]. Unlike other markers which do not acquire chamber-specific expression until the maturation of the heart, both MLC-2 isoforms are known to coexpressed in the initial stages in mouse cardiogenesis and then localise to specific cardiac chambers as early as the stage of cardiac septation [18]. MLC-2v gene expression increases around 10-fold during the ventricular loop formation and septation [22,23]. On the other hand, MLC-2a mRNA levels are high in the outflow tract and negligible in the inflow tract [21]. Thus, MLC-2 isoforms serve as useful markers to track and study the process of cardiogenesis [21].

Even though the sequence of developmental milestones in cardiogenesis remains largely similar in vertebrates, the expression dynamics of RLCs are variable among different species [24,25,26,27,28,29,30]. Cardiac development during murine embryogenesis is evident as early as embryonic day 8 (E8) and continues until after birth [31]. In murine embryogenesis, MLC-2v gene localisation is observed in the early stages (E8 onwards), while most of the markers appear in the late stages of murine cardiogenesis (e.g., post-septation) [32]. The expression of MLC-2v gene is restricted to the ventricular segment of the primitive heart tube in murine embryogenesis [33]. In loss-of-function studies where MLC-2v gene in mice embryos is completely knocked out, ventricular dysfunction occurs at E11.5 [34] and embryos die around E12.5 [35]. It is interesting to note that MLC-2a protein levels increase to levels comparable to those of MLC-2v proteins in the ventricles of the knockout group on E12. Even though this seems to suggest a compensatory effect, the substitution causes structural defects and compromises cardiac contractility [35]. This emphasises the importance of local isoform gene expression. As the heart becomes fully matured, the MLC-2v is exclusively expressed in the left ventricular myocardium [32].

In situ hybridisation experiments have shown that MLC-2a gene is clearly expressed in the early stages (E8) of murine embryogenesis [36,37]. By E12.5, the expression of MLC-2a gene is downregulated in the ventricular chambers in mice [35] and largely restricted to the atrial chambers [18]. Thus, MLC-2a can serve as a negative marker for ventricular chamber specification. On the other hand, in vitro experiments (embryonic stem cell cardiogenesis) show that MLC-2a gene is detectable from day 6 while MLC-2v gene is expressed from day 9 of differentiation. The fact that the expression of both the genes occurs at an early stage suggests that expression may be physiologically independent and affected by other factors (e.g., structural/positional) instead [32]. On that note, MLC-2a knockout mouse models display abnormal structural development (e.g., enlarged and less defined heart tubes) during cardiogenesis and impaired contractility and experience lethality at E10.5–11.5 [38]. As MLC-2a is exclusively expressed in the myocardium (not the vessels), this mouse model also proves that baseline cardiac function is necessary for optimal angiogenesis during embryonic development [38].

Unlike in humans and mice, zebrafish express only one type of MLC-2 gene throughout the heart [20]. The loss of this gene via gene mutation results in the inhibition of myofibrillar assembly, indicating that MLC-2 is necessary for myofibrillogenesis in zebrafish hearts [20]. The loss of MLC-2 results in reduced contractility and cardiomyocyte size [39]. MLC-2 and ventricular myosin heavy chain gene expressed in the heart tube play a key role in the formation of the cardiac cone, conversion of early–medial (E-M) pattern to anterior–posterior (A-P) pattern [40]. Another study used medaka fish to uncover the functions of MLC-2 in cardiogenesis and contraction [41]. Even though it is not explicitly stated that there are no isoforms of MLC-2 gene, only one type of MLC-2 gene (without restrictions to a particular chamber) was studied. It is present throughout cardiogenesis. The knockdown of MLC-2 gene results in large and elongated atrium and large and abnormal sinus venosus [41].

Just as in the murine and zebrafish models, the MLC-2v gene is detected in very early stages (in the anterior part of the cardiac primordia) of avian cardiogenesis and is restricted to the ventricular chambers during development. Again, disruption to MLC-2v gene expression results in abnormal cardiac looping and sarcomeric organisation. The literature seems to suggest there is only one type of MLC-2 gene expressed in birds [42]. From these studies across various species, it is apparent that MLC-2 gene expression is an essential developmental regulator/precursor for proper functions and structural development. Even though the isoform distribution may be different, the stages in which MLC-2 is prevalent are consistent across the different species. The fact that it is expressed at such an early stage (before myogenic precursors) suggests that it may be triggered not physiologically but by other factors such as structural (positional)/molecular cues. The information on longitudinal expression of RLCs has been elucidated largely by in situ hybridisation in cardiogenesis models [21]. Loss-of-function studies via genetic knockout using oligonucleotides or genetic manipulation, such as in retinoid X receptor alpha (RXRα) knockout mouse, are used to study embryonic heart failure displaying ventricular chamber defects [43,44]. Incidentally, RXRa -/- knockout embryos displayed aberrant expression of MLC-2a in thin-walled ventricular chambers at E13.5 while the expression was already downregulated in wild-type counterparts. On the other hand, MLC-2v remained restricted to the chambers in the knockout embryos. Until recently, early cardiogenesis stages were recapitulated in human pluripotent stem cell heart-forming organoids [45]. These organoids are formed with NKX2.5-knockout cells, where NKX2.5 is known to interact with other transcription factors to regulate cardiac structural gene expression and normal cardiac development [46]. While most structural genes maintained comparable levels to those of wild-type organoids, expression of ventricular RLC genes decreased by 1.6-fold in the NKX2.5-knockout organoids [45], suggesting that RLCs play a direct role in heart formation.

## 4. The Role of Regulatory Light Chains in Normal and Diseased Hearts

We have learned that the ablation of ventricular or atrial MLC-2 results in embryonic lethality, emphasising RLCs’ crucial role in cardiac development. However, loss-of-function studies in animal models have impeded the interrogation of RLCs in adult hearts. Instead, molecular and structural studies on RLCs have revealed that phosphorylation of RLC residues is required for proper cardiac muscle contractility in normal and diseased hearts. The RLC is positioned at the IQ motif (lever arm) of the myosin heavy chain molecules [47]. In canine and porcine models, selective removal of RLC from isolated cardiac myosin molecules reduces myosin neck length and causes the myosin head region to become more globular, decreasing the chances of cross-bridge formation [48,49]. Removal of RLC from rabbit skeletal fibres compromises muscle shortening velocity [50,51,52]. Partial extraction (35.2%) of RLC from skinned rabbit skeletal muscle fibres decreases the force of isometric contraction by 7% [50]. Building our knowledge from skeletal muscle fibres, the RLC binding site on skeletal myosin molecule undergoes repetitive conformational (tilt and twist actions) changes during muscle contraction [53]. These data suggest that the RLC’s location on a dynamic part of the myosin molecule may affect striated muscle contraction and maintenance of proper muscle function [5]. There has been only one report of RLC-deprived porcine cardiac myosin, which unexpectedly results in a 2-fold increase in the force of muscle contraction [49]. This result was obtained from an optical-trap-based isometric force in vitro motility assay [49]. Experimental strategies to better characterise RLC dynamics are discussed in a later section.

In humans, there are a number of RLC missense mutations (i.e., R58Q, D166V, E22K, K104E, D94A, A13T, N47K) that are associated with cardiomyopathies [54,55,56,57,58]. The site of mutation affects the severity of disruption to the cardiomyocytes [9]. For example, R58Q and N47K mutations, which are near to the cationic binding site, drastically affect Ca^2+^ binding properties. Mutations at close proximity to the phosphorylation sites, A13T, F18L and P95A, decrease muscle energetics (maximal ATPase activity) [9]. Familial cardiomyopathies commonly present compromised cardiac functions such as diastolic filling abnormalities, systolic dysfunction, decreased ejection fraction and impaired contractility [56,59]. These data show not only that the presence of RLC at its appropriate position is necessary but also that its native form is required for its proper function. To understand the mechanisms underlying cardiac disease progression, transgenic mouse models expressing human RLC carrying the R58Q mutation have been created, reproducing a cardiac hypertrophic phenotype. Mice with the D166V mutation in RLC display myofilament disarray, fibrosis and eventually malignant familial hypertrophic cardiomyopathy [60,61]. Likewise, mice with the D94A mutation experience hypocontractile activity of myosin motors and systolic dysfunction due to dilation of left ventricular chamber, indicating cardiomyopathy [62]. Structural and biochemical studies corroborate the above phenotypic findings. Using polarised fluorescence to measure myosin orientation changes, R58Q mutations in RLC cause the myosin molecule (head and neck region) to be positioned parallel to the thick filaments, thus reducing the availability of myosin heads for cross-bridge formation [56,63]. Cardiac fibres in which R58Q-mutated RLC is exchanged show reductions in force and power of contraction [59]. Indeed, RLCs stabilise the movement of the lever arm of the myosin molecule to ensure normal cardiac function.

One mechanism by which RLCs regulate the movement of the myosin head is via phosphorylation. The phosphorylation of RLCs at their serine residues induces negative charges at the myosin head region. As a result myosin heads are repulsed from the positively charged myosin filament backbone region and move towards the actin filaments (shown by electron microscopic studies), facilitating cross-bridge formation for muscle contraction [64,65,66]. There is also a natural RLC phosphorylation gradient, increasing from endocardium to epicardium, along the transverse axis of the heart [67], observed in human, rodent and rabbit hearts [67,68] via immunohistochemistry. This gradient results in differential tension production and stretch activation response giving rise to cardiac torsion as the heart contracts [67,69]. This torsion (wringing) effect increases ejection fraction. However, this phenomenon has been challenged recently, when quantitative measures using gel electrophoresis reported no spatial gradient of RLC phosphorylation in mice [70,71].

Studies have shown that the baseline phosphorylation levels of ventricular RLCs are about 0.3–0.5 mol Pi/mol RLC (in humans, pigs and rodents) for optimal cardiac function [72]. The RLC phosphorylation also increases Ca^2+^ sensitivity and the rate of cross-bridge kinetics across various species [73,74,75,76]. The phosphorylation of residues increases the affinity between the actin and myosin filaments, facilitating muscle contraction. These activities were captured and elucidated via crystallographic models and fluorescence polarisation spectroscopy [53,77,78].

Structural studies have shown that the phosphorylation of RLCs reduces interfilament spacing (distance between actin and myosin filaments) [75]. This then reduces the time taken for cooperative recruitment of cross-bridges, hence increasing the rate of myocardial force development (enhanced myosin kinetics) [75,79]. Biochemical experimentations show that enhanced RLC phosphorylation levels increase isometric force and peak power output [1,2]. Moreover, transgenic mice with nonphosphorylatable RLCs experience reduced systolic pressure and decreased contractility [80]. Lack of RLC phosphorylation (due to loss of MLCK) in mice results in ventricular hypertrophy and myocyte hypertrophy and eventually leads to compromised cardiac function (e.g., decreased systolic performance) [81].

RLC phosphorylation levels are significantly reduced in heart failure patients [82,83,84]. Animals in which myocardial infarction is induced by left anterior descending coronary artery ligation display a variation in the changes of RLC phosphorylation levels during heart failure progression. In rats, the phosphorylation levels continue to increase at 20 weeks post-MI [85], while rabbits experience a continuous decrease from 2 weeks post-MI [86]. From the various patterns of changes in RLC phosphorylation levels, we and others postulate that the changes may be affected by various factors such as age and heart rates. Here, we summarise the ventricular RLC phosphorylation levels in normal and impaired hearts (Table 1). Some have hypothesised that the increase in RLC phosphorylation levels can be a compensatory effect during heart failure progression [70,87]. Another study also supports that enhanced RLC phosphorylation level does not contribute to hypertrophy but may instead inhibit cardiomyopathy via promoting improved contractile performance [70]. This indicates that RLC phosphorylation can mitigate heart failure progression. Interestingly, RLC phosphorylation can rescue the pathological effects arising from mutated RLCs. Pseudophosphorylation of D166V-mutated RLC prevents abnormal hypertrophy, myofilament disarray and fibrosis [60]. Phosphorylation of R58Q- and A13T-mutated RLC restores Ca^2+^ binding to RLC [88]. Thus, phosphorylated RLCs have potential for treating cardiac diseases and improving patient outcomes.

The mechanisms regulating RLC phosphorylation remain elusive. There are conflicting reports on RLC phosphorylation being initiated by β-adrenergic stimulation or inhibition [91,92]. Nonetheless, it is established that basal cardiac RLC phosphorylation levels are regulated primarily by the balance of activity between cardiac myosin light chain kinase (cMLCK) and myosin light chain phosphatase (MLCP) [71,92,93].

cMLCK is postulated to be a Ca^2+^/calmodulin-dependent kinase and is activated during activation of muscle contraction, namely when cystolic Ca^2+^ ion concentration is high [71]. As the crystallographic structures are unavailable for any MLCK isoforms, it is difficult to elucidate its function in phosphorylating RLCs. Biochemical experiments have shown that smooth muscle and skeletal MLCKs (smMLCK and skMLCK) have Ca^2+^/calmodulin-dependent autoregulatory segments, enabling the binding of RLC to MLCK [94]. As cardiac MLCK contains an autoregulatory segment, it is likely to play a role similar to those of smMLCK and skMLCK. Most studies showed that MLCK is an RLC-dedicated kinase [95] until a recent study showed that human MLCK phosphorylates both RLC and troponin I (TnI) [96]. The following questions remain: (1) What are the roles of other kinases? (2) As MLCK is likely to be Ca^2+^/calmodulin-dependent, are there other kinases involved in maintaining phosphorylation levels in diseased hearts with compromised contraction?

Studies have shown that other kinases are able to phosphorylate RLCs. Zipper-interacting protein kinase (ZIPK) is ubiquitously expressed in the heart. The RLC is a substrate of ZIPK, as identified by unbiased substrate search [97]. ZIPK phosphorylates serine 15 residue and thus is likely to contribute to muscle contraction [98]. However, its role in normal or diseased hearts is not known [71]. Evidence from smooth and nonmuscle cells suggests that ZIPK is activated by upstream signalling pathways (e.g., Ca^2+^ sensitisation, muscle contraction, apoptosis) [99,100]. Other than ZIPK, Rho kinase (ROCK) indirectly regulates RLC phosphorylation levels by binding and activating MLCP [101]. Cardiac RLCs are also phosphorylated by protein kinase C (PKC) via the activation of the α-adrenergic pathway. PKC phosphorylates at sites different from MLCK [102]. CaM-dependent kinase II (CaMKII) phosphorylates RLCs under inotropic conditions [103]. Thus, regulation of enzymatic activity underlying RLC phosphorylation is complex and points to the importance of this process in cardiac regulation.

Increasing RLC phosphorylation may be a promising approach towards improving the cardiac function of diseased hearts and rescuing the functional consequences of RLC mutations. Many studies have reported disparate changes of RLC phosphorylation with heart failure progression in humans or animal models. Possible reasons, including biological disparities (e.g., different origins causing heart failure, age and gender of human patients), should be delved into. Finally, elucidating the complex enzymatic pathways in normal and diseased hearts would open up novel therapeutic avenues based on RLC phosphorylation.

## 5. Experimental Strategies to Study and Exploit the Use of Regulatory Light Chains

The detection and quantification of RLC phosphorylation extracted from cell or tissue samples can be challenging due to the low abundance of phosphoproteins [104]. Various methods have been devised over the years to study and analyse phosphorylation levels of proteins, albeit each with their own pros and cons. For example, mass spectrometry and gel electrophoresis (1D and 2D gel electrophoresis) are critical steps for obtaining phosphoproteomic data but are time-consuming and require careful interpretation [105,106]. Gel electrophoretic methods (e.g., urea–glycerol, Phos-tag and 2D gel electrophoresis) and phospho-specific ELISA are useful for determining phosphorylation levels and hence the kinase activity of targeted proteins. Collective experience suggests that extensive optimisation (e.g., polyacrylamide optimisation, electrophoresis duration) is required to resolve phosphorylated RLCs from different species for repeatable and reliable results. Although phospho-specific ELISA is a promising tool, there are only a few commercially available phospho-specific ELISA kits for targeted proteins. Moreover, the use of phospho-specific ELISA kits and phosphofluro antibodies limits the findings to a particular phosphorylation site rather than the total phosphorylation activity of RLCs [107].

At the moment, there are no imaging modalities to directly visualise RLCs in the sarcomere. Immunohistochemistry (IHC) provides the spatial distribution of RLCs in cells or tissues, electron microscopy enables the movement of myosin heads and actomyosin interactions to be visualised and low-angle X-ray diffraction provides dynamic data on contractile protein motions [108]. Techniques such as electron microscopy, nuclear magnetic resonance (NMR) and X-ray crystallography allow high-resolution models of three-dimensional proteins to be illustrated [109]. However, careful sample preparation, practice and optimisation are necessary [110,111].

Biochemical protein exchange experiment is one of the most powerful experiments involving RLCs because muscle cell contractile function can be monitored before and after the exchange. This technique enables native RLCs in permeabilised muscle fibres to be exchanged with recombinant RLCs with various chemical treatments, primarily involving trifluoperazine (Figure 2). The exchange procedure is well established, as described by Toepfer et al. [1]. Such exchange procedures in an in vitro setting enable the study of the role of RLC isoforms (mutated, phosphorylated, different species, etc.) and the exploration of their therapeutic potential. However, with the current protocol, the exchange efficiency in skinned cardiac muscle fibres is only about 50% [1]. However, there is a need to find the balance between the effectiveness of protein extraction and increasing exchange efficiency as stronger chemicals have the potential to disrupt RLC and other proteins [112]. On the other hand, RLCs exchanged in isolated myosin molecules have an exchange efficiency of more than 85% [113]. The RLC exchange can only be performed on isolated protein solutions, or in permeabilised cells, as the protein does not diffuse through the membrane. Hence, exchanging RLC in vivo is a challenge. If researchers manage to overcome this, the next hurdle will be to target delivery of RLCs to the sarcomere in a specific organ or tissue.

There are useful tools such as in situ hybridisation to locate the longitudinal expression patterns and loss-of-function studies via genetic knockout using oligonucleotides or genetic manipulation to study the role of RLCs in cardiogenesis. As the RLC ventricular isoform is restricted to the ventricles until adulthood, it has the potential to serve as a genetic locus (promoter) for driving ventricle-specific gene expression, as well as a robust marker for ventricular cardiomyocyte selection and enrichment (flow cytometry) and detection (immunohistochemistry) [37,114]. Studies so far use bacterial/viral–RLC promoter-based methods to select ventricular cardiomyocytes from differentiated cells [115,116,117]. However, there are limitations to these techniques, as the use of bacterial or viral components has limited clinical applications due to safety and ethical issues [37,117]. Recent studies have shown the potential to overcome this limitation via the insertion of reporters at an untranslated region of MLC-2, using CRISPR-Cas9 technology [118].

The selection of pure ventricular cardiomyocytes from hiPSC differentiated cells is useful for improving regenerative-medicine-related treatments such as cardiomyocyte cell patches and cell delivery techniques. Even though RLCs are one of the earliest markers that arise in cardiac development, they are used as a late-stage detection marker in iPSC-derived cardiomyocytes, as these are developmentally less mature than human hearts [119,120]. Thus, other markers such as TnI rather than RLC genes can be used to identify cardiomyocytes in the early stages of hiPSC differentiation [119]. Cardiomyocytes derived from patients’ iPSCs can be used to model diseases to uncover the underlying mechanisms resulting in disease progression. For example, cardiomyocytes derived from iPSCs of a patient with R58Q mutation exhibited larger cell size, myofibrillar disarray and irregular beating [121]. There may be a need to improve differentiation protocols, in terms of duration or culture conditions, to promote defined promoter marker expression and gather a more purified cell population [116], for both disease treatment and modelling purposes. Recently, human pluripotent stem cell derived heart-forming organoids recapitulating early cardiogenesis stages were successfully created; they will also be useful for RLC studies [45].

Taken together, the findings related to RLCs are derived from in vitro or in situ conditions, thus far. It will be useful to derive methods to study their activities in an in vivo setting for translation to therapeutic purposes.

## 6. Conclusions

The structure and functional roles of the RLC have been studied for the past five decades [122], but discoveries are still being made that highlight the role of the RLC in myology and embryology and its potential in regenerative medicine. The early expression of cardiac RLCs and their specificity to cardiomyocytes are beneficial for the study of cardiac development and advancements in cardiac regeneration. The unique ability of RLCs to be extracted and re-inserted into the sarcomere by chemical means opens up opportunities for biophysical experimentation to understand RLC function in normal and diseased hearts. Finally, the analeptic effect of phosphorylated RLCs allows us to be optimistic about improvements in the treatment of heart failure. With the current and future research technologies, there are possibilities for more studies to be conducted in physiological settings. Hopefully, RLC-based treatments can be translated from bench to bedside in the near future.

## Figures and Tables

**Figure 1 ijms-22-04351-f001:**
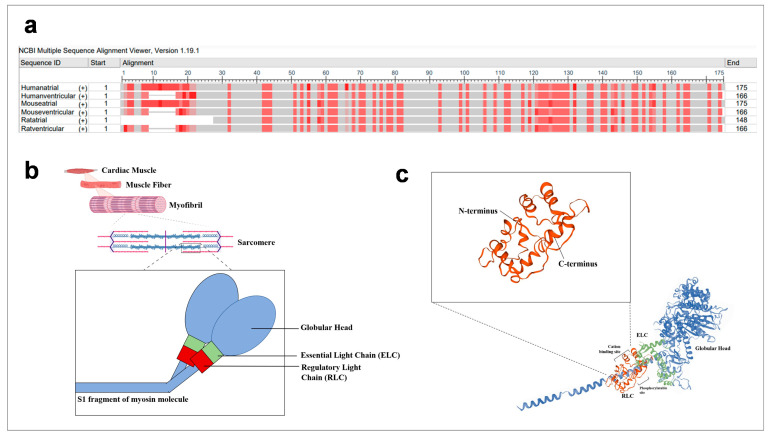
Sequence and illustration of regulatory light chains. (**a**) Atrial and ventricular RLC sequences from human, mouse and rat were aligned with NCBI multiple sequence alignment platform (https://www.ncbi.nlm.nih.gov/tools/msaviewer/ (accessed on 1 March 2021)). Frequency-based differences were used to compare the residue at a position to the column consensus. Darker shades of red indicate greater variation from residues in other rows at that position. (**b**) Spatial illustration of RLC within the sarcomeres of cardiac muscle. The RLC (red component) is located at the lever arm of the S1 fragment of the myosin molecule (blue). (**c**) Schematic representation of key components such as cation binding sites and phosphorylation sites, as well as the interaction of RLC’s N and C termini with myosin heavy chains.

**Figure 2 ijms-22-04351-f002:**
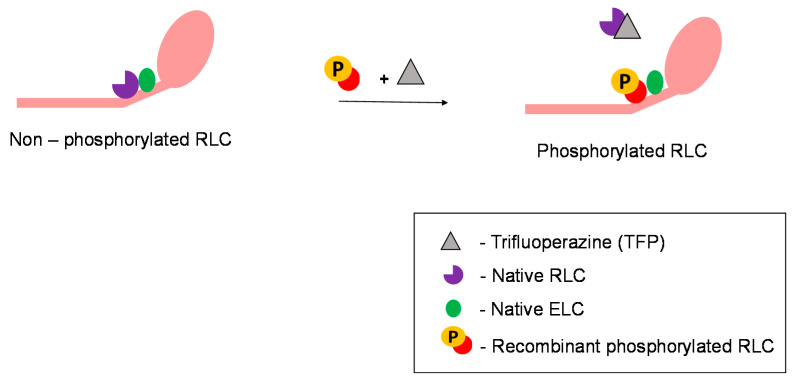
RLC exchange on myosin molecules. Role of trifluoperazine in facilitating RLC exchange.

**Table 1 ijms-22-04351-t001:** The RLC phosphorylation levels in normal and impaired ventricles across varying species. Values are reported with the units mol Pi/mol RLC.

Species	Normal Ventricles (mol Pi/mol RLC)	Diseased Ventricles (mol Pi/mol RLC)	References
Human	0.39–0.40	0–0.6 (End-stage heart failure)	[1,72]
Pigs	0.39	Decrease (Ischaemic failing heart model)	[72,89]
Rat	0.39	0.76 (Ischaemic failing heart model) Decrease (Ischaemic failing heart model)	[1,72,90]
Rabbit	0.36	–	[72]

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
