# Peer review of "Regulatory Light Chains in Cardiac Development and Disease"

_ijms, 2021, doi:10.3390/ijms22094351_

Round 1
Reviewer 1 Report
The review manuscript by Markandran et al. entitled “Regulatory Light Chains in Cardiac Development and Disease” strikes me as timely, well-conceived, well-organized and engagingly written. There are only a few specific points that I feel need addressing as outlined below. There are also a number of grammar mistakes and awkward phrasings that should be fixed to improve readability. Once this is done this paper will make a fine contribution.
Major points
Line 59 is an awkward sentence. I would rewrite it as:
However, the significance of this divalent binding site for regulation of both skeletal and cardiac muscle contraction is not obvious, as the transition in the type of divalent ion at the binding site is slow (you could consider indicating a time constant) on the time-scale of contraction (should be more specific – perhaps time for a twitch??,) signifying that it does not play a primary role in muscle contraction (11).
Furthermore reference 11 doesn’t really address the time scale of contraction nor does references in reference 11 so this concept must have come from somewhere else. Please find a more suitable reference.
Line 102. When introducing development of the heart during murine embryogenesis, it would be useful to orient the reader to the nomenclature used to denote the various stages since this may be unfamiliar to them. An introductory sentence with a suitable reference would take care of this.
Line 391: Reference 119 is missing
Specific suggestions for correcting grammar and improve readability.
Line 36: replace “explored” by “exploited”
Line 38: “highlights knowledge gaps and suggests potential applications”.
Line 46: When “RLC” starts a sentence replace with “The RLC” globally.
Line 54: “also associates with MHC molecules”
Line 65: “in the human cardiac RLC…”
Figure 1 caption: c:” Schematic representation of ley components such as cation binding sites and phosphorylation sites…
Line 86: “The heart is the first organ…”
Line 96: “10-fold”
Line 97” high in the outflow track and negligible in the inflow track….
Line 103 “replace “makers” with “markers”
Line 111 “local isoform gene expression”
Line 120 “The fact that both the genes are present at an early stage suggests…..”
Line 143: “The literature seems to suggest…”
Line 159: “human pluripotent stem cell heart-forming organoids..”
Line 176 “Removal of RLC from rabbit skeletal fibres compromises muscle shortening velocity”
Line 181: “These data suggest…..”
Line 182: “There has been only one report….”
Line 195 “These data show”
Line 199: “Mice with the D166V mutation in the RLC…..”
Line 201: “mice with the D94A mutation”
Line 212” This results in repulsion of myosin heads from the positively….”
Line 222:’ in mice”.
Line 226 “increases the affinity”
Line 257 “have the potential”
Line 293: “Increasing RLC phosphorylation may be a promising approach towards improving cardiac
function of diseased hearts and rescuing the functional consequences of RLC mutations.”
Line 306: “each with there own pros and cons”
Line 311: “Collective experience suggests that extensive optimization…..”
Line 326: “Biochemical protein exchange experiment is one of”
Line 328:” This technique”
Line 329 “RLC’s with various chemical treatments”
Line 355: “efficiency as stronger chemicals have the potential….”
Figure 2 caption “Figure 2: RLC exchange on myosin molecules. Role of Trifluoperazine in facilitating RLC exchange.”
Line 360: “Since the RLC ventricular isoform is restricted…”
Line 380 “differentiation protocols…”
Line 384: “ have been successfully created which will also be useful for RLC studies”
Line 391: “The structure and functional roles of the RLC have been studied…..”
Line 393: “ The early expression of cardiac RLC’s and their specificity to cardiomyocytes…..”
Line 395: “The unique ability of RLC’s ..
Line 398: “phosphorylated RLCs”
Line 400 :” in physiological settings”.
Author Response
Reviewer 1
The review manuscript by Markandran et al. entitled “Regulatory Light Chains in
Cardiac Development and Disease” strikes me as timely, well-conceived, wellorganized and engagingly written. There are only a few specific points that I feel need
addressing as outlined below. There are also a number of grammar mistakes and
awkward phrasings that should be fixed to improve readability. Once this is done this
paper will make a fine contribution.
Thank you for your time and effort in reviewing our manuscript. We appreciate your comments.
Major points
Line 59 is an awkward sentence. I would rewrite it as:
However, the significance of this divalent binding site for regulation of both skeletal
and cardiac muscle contraction is not obvious, as the transition in the type of divalent
ion at the binding site is slow (you could consider indicating a time constant) on the
time-scale of contraction (should be more specific – perhaps time for a twitch??,)
signifying that it does not play a primary role in muscle contraction (11).
Furthermore reference 11 doesn’t really address the time scale of contraction nor
does references in reference 11 so this concept must have come from somewhere
else. Please find a more suitable reference.
Thank you for your kind suggestion. We have made the following changes from Line 59, and
added the time component to better explain the phenomenon of contraction.
“The significance of this divalent binding site for regulation of both skeletal and cardiac muscle
contraction was rejected as the transition in the type of divalent ion at the binding site is slow
(~8s-1
) on the time-scale of contraction (~50ms for twitch contractions) (9, 11, 12). This
conclusion is gathered based on in vitro experiments (9). However, loss-of-function studies
via site directed mutagenesis, showed that the divalent binding site is necessary for muscle
contraction (13, 14).”
We found that references (11) and (12) contain the information of time scales, hence they
have been cited in text.
11. Janssen PML. Kinetics of cardiac muscle contraction and relaxation are linked and
determined by properties of the cardiac sarcomere. American journal of physiology Heart and
circulatory physiology. 2010;299(4):H1092-H9.
12. Grabarek Z. Insights into modulation of calcium signaling by magnesium in calmodulin,
troponin C and related EF-hand proteins. Biochim Biophys Acta. 2011;1813(5):913-21.
Line 102. When introducing development of the heart during murine embryogenesis, it
would be useful to orient the reader to the nomenclature used to denote the various
stages since this may be unfamiliar to them. An introductory sentence with a suitable
reference would take care of this.
We have added this to Line 105.
“Cardiac development during murine embryogenesis is evident as early as embryonic day 8
(E8) and continues until after birth (31).”
Related reference:
31. Savolainen SM, Foley JF, Elmore SA. Histology atlas of the developing mouse heart
with emphasis on E11.5 to E18.5. Toxicol Pathol. 2009;37(4):395-414.
Line 391: Reference 119 is missing
We have cited the reference on Line 398.
Related reference:
123. Perrie WT, Perry SV. An electrophoretic study of the low-molecular-weight
components of myosin. The Biochemical journal. 1970;119(1):31-8.
Specific suggestions for correcting grammar and improve readability.
Line 36: replace “explored” by “exploited”
Line 38: “highlights knowledge gaps and suggests potential applications”.
Line 46: When “RLC” starts a sentence replace with “The RLC” globally.
Line 54: “also associates with MHC molecules”
Line 65: “in the human cardiac RLC…”
Figure 1 caption: c:” Schematic representation of ley components such as cation
binding sites and phosphorylation sites…
Line 86: “The heart is the first organ…”
Line 96: “10-fold”
Line 97” high in the outflow track and negligible in the inflow track….
Line 103 “replace “makers” with “markers”
Line 111 “local isoform gene expression”
Line 120 “The fact that both the genes are present at an early stage suggests…..”
Line 143: “The literature seems to suggest…”
Line 159: “human pluripotent stem cell heart-forming organoids..”
Line 176 “Removal of RLC from rabbit skeletal fibres compromises muscle shortening
velocity”
Line 181: “These data suggest…..”
Line 182: “There has been only one report….”
Line 195 “These data show”
Line 199: “Mice with the D166V mutation in the RLC…..”
Line 201: “mice with the D94A mutation”
Line 212” This results in repulsion of myosin heads from the positively….”
Line 222:’ in mice”.
Line 226 “increases the affinity”
Line 257 “have the potential”
Line 293: “Increasing RLC phosphorylation may be a promising approach
towards improving cardiac
function of diseased hearts and rescuing the functional consequences of RLC
mutations.”
Line 306: “each with there own pros and cons”
Line 311: “Collective experience suggests that extensive optimization…..”
Line 326: “Biochemical protein exchange experiment is one of”
Line 328:” This technique”
Line 329 “RLC’s with various chemical treatments”
Line 355: “efficiency as stronger chemicals have the potential….”
Figure 2 caption “Figure 2: RLC exchange on myosin molecules. Role of
Trifluoperazine in facilitating RLC exchange.”
Line 360: “Since the RLC ventricular isoform is restricted…”
Line 380 “differentiation protocols…”
Line 384: “ have been successfully created which will also be useful for RLC studies”
Line 391: “The structure and functional roles of the RLC have been studied…..”
Line 393: “ The early expression of cardiac RLC’s and their specificity to
cardiomyocytes…..”
Line 395: “The unique ability of RLC’s ..
Line 398: “phosphorylated RLCs”
Line 400 :” in physiological settings”.
Thank you very much for correcting our grammar, spelling and sentence structure. We have
made the suggested changes in our manuscript.
Reviewer 2 Report
The review is a very nice piece of work summarizing current knowledge about structure and function of myosin regulatory light chains (RLCs) in health and disease, including a brief discursion on methodological approaches exploiting RLC functions in cell biological and pathophysiological aspects e.g. to study cardiogenesis, for usage as early disease markers, or therapeutic purposes.
The review is comprehensively written, focuses on RLCs in cardiac muscle function and most noteworthy, it critically discusses yet poorly understood aspects providing some interesting suggestions for future directions in the field.
There is nothing special to criticize, the review merits publication!
Author Response
Reviewer 2
The review is a very nice piece of work summarizing current knowledge about structure
and function of myosin regulatory light chains (RLCs) in health and disease, including
a brief discursion on methodological approaches exploiting RLC functions in cell
biological and pathophysiological aspects e.g. to study cardiogenesis, for usage as
early disease markers, or therapeutic purposes.
The review is comprehensively written, focuses on RLCs in cardiac muscle function
and most noteworthy, it critically discusses yet poorly understood aspects providing
some interesting suggestions for future directions in the field.
There is nothing special to criticize, the review merits publication!
Thank you for your time and effort in reviewing our manuscript. We appreciate your comments.